# In Vitro-Produced Equine Blastocysts Exhibit Greater Dispersal and Intermingling of Inner Cell Mass Cells than In Vivo Embryos

**DOI:** 10.3390/ijms24119619

**Published:** 2023-06-01

**Authors:** Muhammad Umair, Veronica Flores da Cunha Scheeren, Mabel M. Beitsma, Silvia Colleoni, Cesare Galli, Giovanna Lazzari, Marta de Ruijter-Villani, Tom A. E. Stout, Anthony Claes

**Affiliations:** 1Department of Clinical Sciences, Faculty of Veterinary Medicine, Utrecht University, 3584 CM Utrecht, The Netherlandsm.villani@uu.nl (M.d.R.-V.);; 2Avantea srl, Via Porcellasco 7/F, 26100 Cremona, Italy

**Keywords:** equine, blastocysts, in vitro embryo production, ICM, cell lineage segregation

## Abstract

In vitro production (IVP) of equine embryos is increasingly popular in clinical practice but suffers from higher incidences of early embryonic loss and monozygotic twin development than transfer of in vivo derived (IVD) embryos. Early embryo development is classically characterized by two cell fate decisions: (1) first, trophectoderm (TE) cells differentiate from inner cell mass (ICM); (2) second, the ICM segregates into epiblast (EPI) and primitive endoderm (PE). This study examined the influence of embryo type (IVD versus IVP), developmental stage or speed, and culture environment (in vitro versus in vivo) on the expression of the cell lineage markers, CDX-2 (TE), SOX-2 (EPI) and GATA-6 (PE). The numbers and distribution of cells expressing the three lineage markers were evaluated in day 7 IVD early blastocysts (*n* = 3) and blastocysts (*n* = 3), and in IVP embryos first identified as blastocysts after 7 (fast development, *n* = 5) or 9 (slow development, *n* = 9) days. Furthermore, day 7 IVP blastocysts were examined after additional culture for 2 days either in vitro (*n* = 5) or in vivo (after transfer into recipient mares, *n* = 3). In IVD early blastocysts, SOX-2 positive cells were encircled by GATA-6 positive cells in the ICM, with SOX-2 co-expression in some presumed PE cells. In IVD blastocysts, SOX-2 expression was exclusive to the compacted presumptive EPI, while GATA-6 and CDX-2 expression were consistent with PE and TE specification, respectively. In IVP blastocysts, SOX-2 and GATA-6 positive cells were intermingled and relatively dispersed, and co-expression of SOX-2 or GATA-6 was evident in some CDX-2 positive TE cells. IVP blastocysts had lower TE and total cell numbers than IVD blastocysts and displayed larger mean inter-EPI cell distances; these features were more pronounced in slower-developing IVP blastocysts. Transferring IVP blastocysts into recipient mares led to the compaction of SOX-2 positive cells into a presumptive EPI, whereas extended in vitro culture did not. In conclusion, IVP equine embryos have a poorly compacted ICM with intermingled EPI and PE cells; features accentuated in slowly developing embryos but remedied by transfer to a recipient mare.

## 1. Introduction

Coincident with blastocyst formation, a mammalian embryo undergoes two cell lineage segregations; the first is characterized by differentiation of the outer cells into trophectoderm (TE), which form the outer cell layer of the blastocyst and are subsequently restricted to the placenta, whereas the remaining cells compact into the inner cell mass (ICM) [1,2]. During the second cell lineage segregation, the ICM differentiates into the epiblast (EPI) and primitive endoderm (PE). The pluripotent cells of the EPI give rise to the embryo proper, whereas the PE cells migrate to line the inside of the trophectoderm, thereby completing the formation of the yolk sac, the bilaminar primitive nutrient absorbing structure (primitive placenta) [1,2]. The process of cell lineage segregation in embryos involves differential expression of transcriptional regulators and, as a result, the different cell types, e.g., TE, EPI, and PE, can be distinguished by differential expression of transcription factors; for example, TE cells express CDX-2 (caudal-type homeodomain protein) and GATA-3 (GATA-3 binding protein), whereas EPI cells specifically express SOX-2 ((SRY (Sex Determining Region Y) -box 2)) and NANOG (homeobox protein nanog) and PE cells express GATA-6 (GATA-6 binding protein) [3]. Although all mammalian embryos are thought to undergo the same two cell lineage segregations, and the roles of some critical transcription factors appear to be conserved [4], there are species-specific differences in the roles of certain transcription factors, the spatial pattern of cell lineage specification, and in the developmental stage at which cell lineage segregation takes place; for example, it occurs at an earlier embryonic age in mice (day 4.5, [5]) than in human (day 7, [6]), cattle (day 8, [7]), or pig (day 7, [8]) embryos. The process of cell lineage segregation has not been studied in depth in equine embryos after in vivo development, although Enders [9] suggested that PE cells migrated directly to line the TE at around day 8 without first participating in the formation of an obvious ICM. Aspects of cell lineage segregation have been examined more extensively in the context of the pluripotency of embryonic stem cells [10,11] or for in vitro produced (IVP) embryos, although these studies either lacked a marker to positively identify the pluripotent EPI cells [12] or focused almost entirely on EPI specification (using SOX-2) [13]. It has been shown that the pluripotency marker OCT-4/POU5F1 is unreliable for detecting equine EPI cells because it is not solely restricted to the ICM of horse embryos [11,12]. Furthermore, SOX-2 appears to be a better marker than OCT-4 because it specifically stains the EPI in IVP horse embryos [13]; CDX-2 and GATA-6 seem to be well conserved across species [4]. To better investigate possible differences in the timing and spatial distribution of cell lineage segregation between equine IVP and in vivo derived (IVD) embryos, it is imperative to use markers for each of the three cell lineages simultaneously.

Although the likelihood of pregnancy after the transfer of fresh [14] or frozen-thawed [15] equine IVP blastocysts exceeds 70%, this is approximately 15 to 20% lower than after the transfer of fresh IVD blastocysts [16], suggesting that the developmental competence of IVP embryos is slightly lower than that of embryos that develop in vivo. In addition, the speed of in vitro embryo development affects embryo quality since day 7 or day 8 IVP blastocysts are more likely to yield a viable pregnancy than day 9 IVP blastocysts [15]. Although more slowly developing IVP embryos are, therefore, likely to be compromised in some way, it is not yet clear what the most important aberrations are [17]. In fact, assessing the quality of IVP equine embryos is challenging because they do not have a grossly visible blastocyst cavity, and the ICM is not readily appreciated [18]. However, it is possible that, while slowly developing IVP embryos do not have lower total cell numbers [17], they may have reduced numbers of cells in the ICM or EPI and, as a result, may fail to develop an embryo proper; this could contribute to lower pregnancy and higher early embryonic loss rates. Furthermore, even though in vitro culture conditions for equine embryos have improved markedly over time, as indicated by increased blastocyst production rates in clinical programs [19], they are still unable to completely mimic in vivo conditions. The importance of the in vivo environment for ‘normal’ specification of the different cell lineages was indicated by the loss of the unexpected OCT-4/POU5F1 staining of the TE cells of equine IVP embryos following a 2–3 period in the uterus of a mare [20]. In short, the effects of in vitro culture on the timing and pattern of early equine embryo cell fate decisions have not been examined in detail, despite their potential implications for subsequent embryo developmental competence. The aim of this study was to examine the influence of the conditions in which the embryo developed (IVD versus IVP), developmental stage (early blastocyst versus blastocyst: IVD), speed of in vitro embryo development (fast versus slow development: IVP) and environment (in vitro versus in vivo) for extended culture of IVP blastocysts on the pattern of expression of markers for the three cell lineages (CDX-2 for TE; SOX-2 for EPI; and GATA-6 for PE) in equine embryos.

## 2. Results

### 2.1. Expression of Cell Lineage Markers in IVD Early Blastocysts (n = 3) and Blastocysts (n = 3)

CDX-2 expression was exclusive to the TE of both IVD early blastocysts (Figure 1(A1)) and blastocysts (Figure 1(B1)), with no expression evident in cells positioned within the light microscopically visible ICM. SOX-2 was expressed not only in the ICM but was also co-expressed with CDX-2 in the TE of early blastocysts (Figure 1(A2)). This co-expression of SOX-2 in CDX-2 positive TE cells was almost completely lost in later blastocysts. In the later blastocysts, SOX-2 was expressed strongly in the presumed EPI (Figure 1(B2,B4)). GATA-6 expression was visible in cells within the ICM of early blastocysts and also co-expressed with SOX-2 at the periphery of the ICM (Figure 1(A3,A4)). This co-expression of SOX-2 and GATA-6 was no longer evident in later blastocysts, where GATA-6 was expressed exclusively in the PE (Figure 1(B3,B4)) and SOX-2 expression was only detected in the EPI (Figure 1(B2,B4)). In IVD blastocysts, the PE was evident as a single cell layer lining the inner surface of the TE (Figure 1(B3,B4)).

### 2.2. Expression of Cell Lineage Markers in IVP Day 7 Blastocysts (n = 5)

CDX-2 was consistently expressed only in the TE of IVP blastocysts. (Figure 2(A1–D1)). By contrast, SOX-2 expression varied between IVP blastocysts with regard to exact distribution (Figure 2(A2,C2)) and the number of SOX-2 positive cells (Figure 2(A4–D4)). Indeed, SOX-2-positive cells were scattered around the entire ICM, which extended over approximately half of the central ‘cavity’ of IVP blastocysts. In addition, SOX-2 was co-expressed either weakly (Figure 2(B2–D2)) or strongly (Figure 2(A2)) with CDX-2 in the TE of three of five day 7 IVP blastocysts, although it was restricted to the presumptive EPI in the other two (Figure 2(B2,B4,C2,C4)). The distribution of GATA-6 expression was also variable in IVP blastocysts, although it was restricted to the PE of one of 5 embryos (Figure 2(B3,B4,C3,C4)), there was some co-expression of GATA-6 and CDX-2 in the TE of the remaining four (Figure 2(A3,C3,D3)).

### 2.3. Differences in Embryo Size, Total Cell Number and TE, PE, and EPI Allocation between Day 7 IVD (n = 3) and IVP (n = 5) Blastocysts

Although embryo size did not differ markedly between day 7 IVD (210 ± 19 µm) and IVP blastocysts (184 ± 9 µm), the total cell number was significantly higher in the IVD blastocysts (486 ± 81 versus 317 ± 21; IVD versus IVP) (Figure 3A). At the same time, the total number of TE cells was also higher in IVD (395 ± 54) than in IVP blastocysts (264 ± 34) (Figure 3B); the percentage of cells classified as TE did not differ between them (82 ± 3% and 83 ± 5%, respectively). In contrast, neither the number nor the percentage of PE cells differed between IVD (52 ± 23, 10 ± 3%; respectively) and IVP (38 ± 8, 12 ± 3%) blastocysts. Similarly, while the number of EPI cells tended (*p* = 0.05) to be higher in IVD (34 ± 10) than in IVP blastocysts (14 ± 7), the percentages of EPI cells did not differ significantly (7 ± 2% and 4 ± 2%, respectively). On the other hand, the mean inter-epiblast cell distance was significantly higher in IVP blastocysts (52 ± 6 µm) than in IVD blastocysts (35 ± 3 µm) (Figure 3C).

### 2.4. Differences between Day 7 (Fast Developing, n = 5) and Day 9 (Slow Developing, n = 9) IVP Blastocysts

As previously shown for day 7 IVP blastocysts, CDX-2 expression was exclusive to the TE of IVP embryos that only reached the blastocyst stage after 9 days (Figure 4(A1–D1)). The SOX-2 (Figure 4(C2,D2)) and GATA-6 (Figure 4(C3,D3)) positive cells in day 9 IVP blastocysts were spread over an even wider area than in day 7 IVP blastocysts (Figure 4(A2,B2) and Figure 4(A3,B3), respectively). SOX-2 or GATA-6 were co-expressed with CDX-2 in the TE of all day 9 IVP blastocysts (5/9 and 9/9, respectively: Figure 4(C2,C4,D4)), and most day 7 IVP blastocysts (3/5 and 4/5, respectively: Figure 4(A2,A4)). There were no significant differences in embryo size (184 ± 9 vs. 200 ± 31 µm), total cell number (317 ± 31 vs. 377 ± 104), total TE cell number (264 ± 34 vs. 285 ± 101), or total EPI cell number (14 ± 7 vs. 18 ± 6) between day 7 (fast-developing) and day 9 (slow developing) IVP blastocysts. Similarly, there were no differences in the percentages of TE (83 ± 5% vs. 75 ± 10%) and EPI (4 ± 2% vs. 6 ± 3%) cells between day 7 (fast developing) and day 9 (slow developing) blastocysts. By contrast, the total PE cell number was significantly higher in day 9 (71 ± 33) than in day 7 (38 ± 9) IVP blastocysts (Figure 5A), although the percentage of PE cells did not differ between them (19 ± 10% vs. 12 ± 3%, respectively). However, the inter-epiblast cell distance was higher in day 9 (68 ± 9 µm, slow developing) than in day 7 (52 ± 6 µm, fast-developing) IVP blastocysts (Figure 5A).

### 2.5. Effect of Two Days Extra Culture of Day 7 IVP Blastocysts In Vitro (n = 5) or In Vivo (n = 3)

Additional culture did not affect the expression of CDX-2, which remained restricted to the TE (Figure 6(A1–E1)). However, whereas 2 days of additional culture in vitro had no effect on the distribution of SOX-2 positive cells (scattered: Figure 6(A2,B2)), after 2 days in the uterus of a mare the SOX-2 cells had compacted in two out of three embryos (Figure 6(C2,D2)). In the third IVP embryo, collected 2 days after transfer to a mare’s uterus, no SOX-2 positive cells were detected (Figure 6(E2)). After 2 days of additional in vitro culture, SOX-2 positive cells were still present in the TE, whereas SOX-2 staining had disappeared from the TE after 2 days of in vivo culture. As with SOX-2, GATA-6 expression was still present in the TE of day 7 IVP blastocysts cultured for an additional 2 days in vitro (Figure 6(A3,B3)). After 2 days of in vivo development, the distribution of GATA-6 expression was variable (Figure 6(C3,D3)), but it was less evident in TE cells. Embryo size (404 ± 20 µm), total cell number (997 ± 77), total TE cell number (827 ± 47), and total PE cell number (103 ± 11) were significantly higher in IVP embryos collected 2 days after transfer to a mare’s uterus than in IVP embryos cultured in vitro for 2 days (186 ± 26 µm, 350 ± 133, 265 ± 100 and 16 ± 9, respectively). Total EPI cell number tended to be higher (*p* = 0.09) in IVP embryos after 2 days in a mare’s uterus than in an in vitro culture (40 ± 13 vs. 11 ± 6, respectively).

## 3. Discussion

This study compared the expression of markers for the first three cell lineages in IVD and IVP equine embryos. In IVD horse early blastocysts, the first cell lineage segregation (TE versus ICM) had been completed, and the second cell lineage differentiation (EPI versus PE) had been initiated. By the blastocyst stage, the second cell lineage decision had also been completed, and the EPI cells were much more compacted than at the early blastocyst stage. In comparison, IVP blastocysts lagged behind in development, as evidenced by a lower cell number, a less compacted ICM, and, most obviously, because the second lineage differentiation was not complete in 3/5 of the day 7 IVP blastocysts. Moreover, the EPI cells in IVP blastocysts were intermingled with PE cells in a poorly compacted ICM, reminiscent of the early stages of EPI:PE segregation in other species [21]. In IVP embryos that were slow to reach the blastocyst stage (9 rather than 7 days), compaction of the ICM and EPI was even more delayed. During extended culture, the environment (in vivo versus in vitro) had a marked impact on subsequent embryo development and cell differentiation. Day 7 IVP blastocysts cultured in vitro for 2 more days showed little or no progress in EPI organization and compaction, whereas IVP blastocysts transferred to the uterus of a synchronized mare for 2 days were more developed (higher total cell number) and showed completion (or failure in one case) of the second cell lineage segregation (organization and compaction of EPI cells). These observations extend those of Choi et al. [12], although including a positive marker for EPI cells did allow a subtle refinement of their conclusion that the EPI and PE cells in equine embryos do not first aggregate together to form an ICM. In our study, in both IVD early blastocysts and IVP blastocysts, EPI and PE cells clustered together in an apparent ICM before the PE migrated to line the rest of the blastocoele cavity; the difference presumably arising in part from subtle differences in embryo developmental stage at the time of staining.

In mouse embryos, the first cell lineage segregation is considered to be completed by the morula stage, when CDX-2 is expressed solely by the TE [22]. In horse embryos, CDX-2 expression was already restricted to the TE of IVD early blastocysts, indicating that the first cell lineage segregation is completed at an earlier stage, e.g., the morula stage, and possibly prior to descent into the uterus. The second cell lineage division had not been completed in IVD early blastocysts, as indicated by the co-expression of SOX-2 and GATA-6 at the periphery of the ICM. In IVD early blastocysts, SOX-2 was co-expressed with CDX-2 in the TE, a phenomenon also reported in mouse early blastocysts, where it was suggested that SOX-2 co-expression performs a role in the establishment of the TE lineage [23,24]. Indeed, knocking out SOX-2 in two-cell stage mouse embryos caused embryonic arrest characterized by the absence of CDX-2 expression [23]. In this respect, SOX-2 might also perform a role in initial TE specification during the early development of horse embryos.

Early differentiation and separation of EPI and PE were observed in in vivo horse early blastocysts in that GATA-6 positive cells encircled the SOX-2 positive cells. This cellular organization contrasted to mouse early blastocysts, in which EPI cells are intermingled with PE cells, often referred to as a ‘salt and pepper’ expression pattern [25]; the intermingling of PE and EPI we observed in IVP equine blastocysts therefore, either emphasizes that they are at an earlier stage of development, or suggests a different (aberrant) spatial pattern of the second cell fate decision. In horse IVD blastocysts, the second cell lineage had been completed since the three cell lineage markers (CDX-2, GATA-6, and SOX-2) were expressed separately in the presumptive TE, PE, and EPI cells, respectively. This distinct expression of cell lineage markers in TE, PE, and EPI at the blastocyst stage is similar to other species, including man, cow, and pig [26], but is different to mouse [23,27] and goat [28,29] blastocysts (co-expression of SOX-2 with CDX-2 in the TE). Interestingly, a single cell layer of PE could be seen to line the inner surface of the TE, completing the yolk sac, as early as day 7 in IVD equine blastocysts, slightly earlier than previously observed by electron microscopy [9] and earlier than described in other large animal species including man, cow, and pig (day 10) [4,26].

The dispersal of EPI cells, measurable as a greater mean inter-EPI cell distance, observed in IVP compared to IVD equine blastocysts (although total and relative EPI cell number were not different) suggests an impaired, or delayed, organization and compaction of the EPI in IVP blastocysts. This delay in EPI compaction might contribute to the reduced viability and developmental competence of IVP embryos compared to IVD embryos, as evidenced by the 15 to 20% lower pregnancy rate after transfer [30]. One possible impact of delayed EPI compaction may have been indicated by the failure to find any SOX-2 positive cells in one of the 3 IVP embryos recovered 2 days after transfer to a mare’s uterus; as will be discussed below, this embryo would almost certainly be non-viable. Similarly, IVP bovine blastocysts have been reported to suffer a higher incidence of defective embryonic lineage specification (particularly affecting the EPI), which was proposed to contribute to the 10–40% lower likelihood of pregnancy after transferring IVP than IVD bovine embryos to recipient cows [31,32,33].

In addition to in vitro production per se, the second cell lineage segregation appears to be influenced by, or aberration in this segregation may contribute to, a slower rate of in vitro development because the SOX-2 positive cells in blastocysts that took 9 days to reach the blastocyst stage were even more scattered and the EPI less compact (higher inter-EPI cell distance) than in embryos that reached the blastocyst stage in 7 days. Even though the total and relative number of SOX-2 positive cells did not differ between day 9 (slow developing) and day 7 (fast developing) IVP blastocysts, it is possible that a delay in accumulating or specifying sufficient ICM and EPI cells, or indeed of specifying TE cells, explains the delayed formation of an appreciable blastocoele and TE layer, and contributes to the lower pregnancy rates after the transfer of slowly developing (day 9 or later) IVP blastocysts [15]. Moreover, it is tempting to speculate that the dispersed and uncompacted nature of the EPI in IVP blastocysts may predispose to the splitting of the EPI cells into two separate populations within the embryo to form two separate EPIs. If so, this could explain the heightened risk of monochorionic-monozygotic twins observed after the transfer of IVP blastocysts (approximately 1%) [34], a phenomenon that is encountered only rarely for embryos that develop to the blastocyst stage in vivo [35].

The importance of (micro)environmental cues for equine embryonic cell lineage specification was demonstrated by the marked effects of culturing IVP embryos for an additional 2 days in a mare’s uterus, compared to a Petri dish. Day 7 IVP blastocysts cultured in vivo for 2 days showed thinning of the zona, development of a capsule, and a marked increase in total cell number (997 ± 77) accompanied by compaction of the ICM and/or EPI. By contrast, IVP blastocysts cultured for an additional 2 days in vitro showed few signs of further development, with only a modest increase in total cell number, no obvious compaction of the EPI and incomplete spatial segregation of SOX-2 and GATA-6 positive cells in the ICM. In this regard, uterine factors which promote embryo development and differentiation events, including EPI compaction, are presumably absent in in vitro culture. In the cow, embryokines, such as insulin-like growth factor 1 and colony-stimulating factor 2, have been proposed to be key uterine factors that modulate and promote the development of the embryo to later blastocyst stages [36] and may perform a role in EPI specification and organization. In this respect, it would be interesting to investigate whether co-culture with endometrial cells, tissues, or fluids or supplementation of in vitro culture systems with putative embryokines would improve the in vitro development of equine embryos. Interestingly, as mentioned previously, SOX-2 expression was completely absent in one of the day 7 IVP blastocysts cultured in vivo for 2 days, although expression of CDX-2 and GATA-6 was present. The authors hypothesize that this is an IVP embryo that has failed to specify an EPI. Transfer of such an embryo into the uterus of a mare could potentially result in a pregnancy, but presumably, a vesicle that would fail to form an embryo proper, i.e., an anembryonic vesicle [37]. In this respect, a significant percentage (26%) of OPU-ICSI pregnancies lost after the first positive pregnancy scan are lost before the formation of the embryo proper and/or recorded as anembryonic vesicles [38].

## 4. Materials and Methods

### 4.1. In Vivo Derived Embryos: Collection, Initial Assessment, and Fixing

Animal procedures were approved by Utrecht University’s Animal Experimentation Committee (permit number: 1080020185164). In vivo embryos (*n* = 6) were recovered by flushing the uterus of donor mares with lactated Ringer’s solution (Baxter Nederland BV, Utrecht, the Netherlands), as described by Stout [39], 9 days after induction of ovulation with buserelin acetate (0.33 μg/kg Suprefact^®^, IM: CHEPLAPHARM, Greifswald, Germany). One day after induction of ovulation, when the mares were in estrus with a large preovulatory follicle, they had been inseminated with semen from a fertile stallion; ovulation was confirmed by the emptying of the preovulatory follicle and replacement by a corpus hemorrhagicum which was detected by transrectal ultrasonography on the day after insemination. Recovered embryos were classified as early blastocysts (*n* = 3) or blastocysts (*n* = 3), as described by McCue [40]; early blastocysts had a thick zona pellucida, small blastocoel cavity, and a barely visible capsule, whereas blastocysts had a large blastocoel, a distinct ICM and a clearly discernible capsule between the trophectoderm and an attenuated zona pellucida. All embryos were fixed in 2% paraformaldehyde for 30 min at room temperature (RT; 19–21 °C), washed twice in PBST (PBS containing 0.1% Triton X100; Sigma-Aldrich, Zwijndrecht, the Netherlands) for 5 min, then stored in PBST at 4 °C until immunostaining.

### 4.2. In Vitro Produced Embryos: Production, Culture, and Fixing

In vitro embryos were produced as described by Lazzari et al. [41]. Briefly, immature oocytes were collected by transvaginal aspiration of antral follicles ≥ 5 mm or by scraping follicles from post-mortem ovaries, and shipped in H-SOF (HEPES buffered synthetic oviductal fluid [42]) to an assisted reproduction laboratory for in vitro maturation, intracytoplasmic sperm injection, and in vitro culture to the blastocyst stage. The time required (day of ICSI = day 0) for in vitro embryos to reach the blastocyst stage was recorded. All IVP blastocysts were slow-frozen and thawed as described by Lazzari et al. [41]; after thawing, embryos were treated with pronase (0.5% in PBS) to remove the zona pellucida before 24 h culture in a modified SOF-IVC medium supplemented with bovine serum albumin (BSA) and amino acids [43], and containing 10% of a mixture (1:1) of fetal calf serum and serum replacement (KnockOut Serum Replacement, Life Technologies, Carlsbad, CA, USA) at 38.5 °C in an atmosphere containing 5% CO_2_ and 5% O_2_. Five day 7 IVP blastocysts (fast development) were fixed to examine the expression of cell lineage markers, as were nine day 9 IVP blastocysts (slow development), to investigate whether the speed of in vitro development was associated with a different pattern of cell lineage marker expression. Additionally, day 7 IVP blastocysts were either cultured in vitro (*n* = 5, without zona pellucida) or in vivo (*n* = 3, with zona pellucida) for two more days. In vitro culture was carried out as described above. For in vivo culture, three day 7 IVP blastocysts were transferred non-surgically into the uterus of recipient mares on day 4 after ovulation and recovered from the uterus by uterine lavage, as described above, 2 days later. All embryos were fixed and stored as described above until further processing.

### 4.3. Immunostaining

Fixed embryos were washed in fresh PBST, then permeabilized by incubating in PBS containing 1% Triton X100 for 1 h at RT. Non-specific binding was blocked by incubating the permeabilized embryos in PBST supplemented with 3% bovine serum albumin (BSA: Sigma-Aldrich) and 5% normal goat serum (NGS; Thermo Fisher Scientific, Carlsbad, CA, USA) for 1 h at RT. Next, the embryos were incubated with solutions of PBST (supplemented with 3% BSA and 5% NGS) containing the primary antibodies (1:500 dilution of mouse monoclonal antibody against CDX-2; M4392A-SUC, BiogeneX, Fremont, CA, USA; and 1:250 dilution of rabbit polyclonal antibody against GATA-6; H-92, Santa Cruz Biotechnology, Inc., Dallas, TX, USA) overnight at 4 °C in a humidified chamber. After washing in PBST supplemented with 3% BSA three times for 10 min, the embryos were incubated with the secondary antibodies (1:250 Alexa Fluor 568-goat anti-mouse, A11031 Mol Probes, Eugene, OR, USA; 1:250 Alexa Fluor 647-goat anti-rabbit, A21244 LifeTech, Carlsbad, CA, USA) in PBST supplemented with 3% BSA and 5% NGS for 2 h at RT. After washing three more times in PBST supplemented with 3% BSA for 10 min, the embryos were incubated with normal mouse IgG_1_ (1:250 SC-3877, Santa Cruz) in PBST supplemented with 3% BSA and 5% NGS for 2 h at RT. Following three 5 min washes as described above, embryos were incubated with SOX-2 (1:250 dilution of mouse monoclonal antibody against SOX-2, E-4, SC-365823 conjugated with Alexa Fluor 488, Santa Cruz) and Hoechst 33342 (1:500, B2261; Sigma-Aldrich) in PBST supplemented with 3% BSA and 5% NGS for 2 h at RT. Finally, the embryos were mounted in a 5 µL droplet of antifade (Vectashield; Vector Laboratories, Newark, CA, USA) on glass slides (Superfrost Plus; Menzel, Braunschweig, Germany).

### 4.4. Confocal Imaging and Image Analysis

A Nikon A1R/STORM confocal microscope (Nikon Instruments Inc., Tokyo, Japan) equipped with four lasers (405 nm, Blue; 488 nm, Green; 561 nm, Red; and 647 nm, Magenta) was used to assess the expression of Hoechst 33342 (all nuclei), SOX-2 (EPI), CDX-2 (TE), and GATA-6 (PE). The dichroic filter was a quad line laser filter (405/488/561/640), and the emission filters were 482/32, 515/30, 595/50, and 700/75 for Hoechst 33342, Alexa-Fluor 488^TM^, Alexa Fluor 568^TM^, and Alexa Fluor 647^TM^, respectively. Z-stacks (1.1 µm thickness with top and bottom acquisition defined manually) were acquired using a 20×/0.75 NA (numerical aperture) air immersion lens (1000 µm working distance). The pinhole size was 26.82 µm, and images (512 × 512 pixels) were collected in the bidirectional mode with a pixel size of 0.65 µm. Image analysis was performed using Fiji Image J2 and Imaris software version 8.2 (BitPlane AG, Zurich, Switzerland). Individual image slices were processed for brightness and contrast using Fiji image J for graphical illustrations. All blastocysts were measured using Imaris (mean of two measurements from trophoblast perimeter to trophoblast perimeter). The total cell number (Hoechst 33342 positive nuclei), and total numbers of presumed TE (CDX-2 positive cells), PE (GATA-6 positive cells), and EPI (SOX-2 positive cells) were also counted using the Imaris object spot detection module. Objects were filtered to contain mean intensities between minimum and maximum intensities (98–1277, 76–119, 70–1156, and 83–201) for blue (Hoechst 33342 positive nuclei), green (SOX-2 positive cells), red (CDX-2 positive cells), and magenta (GATA-6 positive cells) channels, respectively. Mean inter-EPI cell distances were measured using the Imaris spots-to-spots closest distance algorithm.

### 4.5. Statistical Analysis

Shapiro–Wilk tests were used to assess the normality of the data. Differences in embryo size, total cell number, total numbers of TE, EPI, and PE cells, and percentages of all cells classified as TE, EPI, and PE cells were compared between day 7 IVD and IVP embryos using the Mann–Whitney test. Unpaired T-tests with Welch’s correction were used to detect differences in embryo size, total cell number, total numbers of TE, EPI, and PE cells, and percentages of cells classified as TE, EPI, and PE cells between day 7 (fast development) and day 9 (slow development) IVP blastocysts. The Mann–Whitney test was used to detect differences in embryo size, total cell number, and total numbers of TE, EPI, and PE cells between IVP embryos collected 2 days after transfer to a mare’s uterus and IVP embryos cultured in vitro for 2 days. An IVP embryo without SOX-2 positive cells (collected 2 days after transfer to a mare’s uterus) was excluded from the comparison of total EPI cell number. Statistical analysis was performed using GraphPad prism software version 8 (GraphPad Software, San Diego, CA, USA). Differences were considered statistically significant when *p* < 0.05; tendency 0.05–0.09. Data are presented as means ± SD (standard deviation).

## 5. Conclusions

In conclusion, IVP horse blastocysts have a more dispersed EPI than IVD horse blastocysts with an intermingling of EPI and PE cells. The greater dispersion may result from developmental delay and is even more pronounced in slowly developing (day 9) IVP horse blastocysts. However, the EPI of IVP horse blastocysts compacts rapidly after transfer to the uterus of a recipient mare, which indicates that the uterine environment can normalize the developmental differences of IVP embryos. Overall, unraveling and comparing the intra- and extra-embryonic factors that affect the molecular mechanisms underlying cell commitment and the inter-proteomic crosstalk between transcription factors that dictate cell fate, such as CDX-2-, SOX-2-, and GATA-6, seems to be an important aspect of promoting improvement in the efficiency of generating ex vivo-derived high-quality blastocysts in equids and other mammalian species. This might also enhance in vivo developmental competence after intrauterine transfer of IVP-derived embryos propagated by advanced assisted reproductive technologies (ARTs), such as in vitro fertilization (IVF), based either on gamete co-incubation or intracytoplasmic sperm injection (ICSI) [44,45,46,47], and cloning based on somatic cell nuclear transfer (SCNT) [48,49,50].

## Figures and Tables

**Figure 1 ijms-24-09619-f001:**
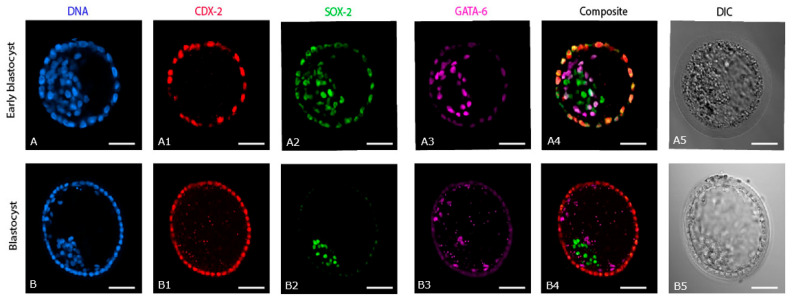
Expression of cell lineage markers (CDX-2, SOX-2, and GATA-6) in in vivo derived early blastocysts and blastocysts. (**A**,**B**) DNA staining of all the nuclei: (**A1**,**B1**) expression of CDX-2, (**A2**,**B2**) expression of SOX-2, (**A3**,**B3**) expression of GATA-6, (**A4**,**B4**) composite of CDX-2, SOX-2 and GATA-6, (**A5**,**B5**) DIC images of the respective early blastocyst and blastocyst. Scale bar = 50 µm.

**Figure 2 ijms-24-09619-f002:**
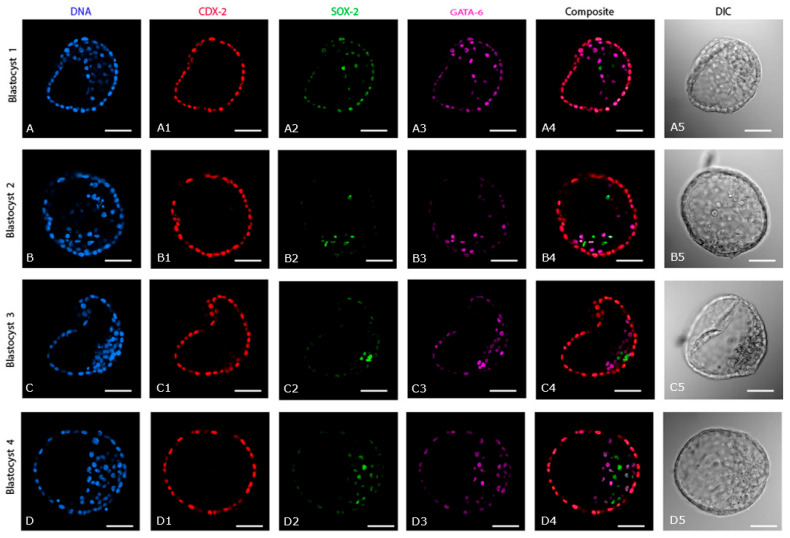
Expression of cell lineage markers (CDX-2, SOX-2, and GATA-6) in day 7 in vitro produced blastocysts. (**A**–**D**) DNA staining of all the nuclei. (**A1**–**D1**) Expression of CDX-2. (**A2**–**D2**) Expression of SOX-2. (**A3**–**D3**) Expression of GATA-6. (**A4**–**D4**) Composite of CDX-2, SOX-2 and GATA-6. (**A5**–**D5**) DIC images of the respective in vitro blastocysts. Scale bar = 50 µm.

**Figure 3 ijms-24-09619-f003:**
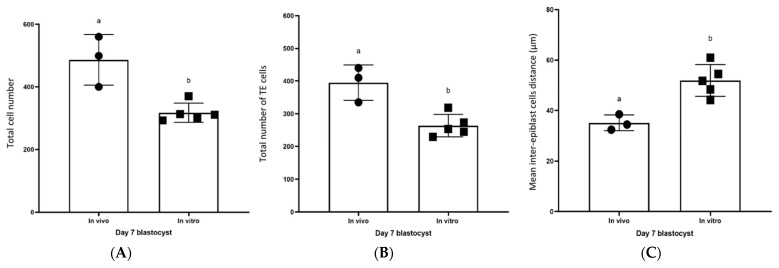
Differences between day 7 IVD and day 7 IVP blastocysts. (**A**) Total cell number, (**B**) Total number of TE cells, (**C**) Mean inter-epiblast cell distance. Individual values are plotted as filled squares or circles; the error bars represent the standard deviation. Columns with different superscripts differ significantly, *p* < 0.05.

**Figure 4 ijms-24-09619-f004:**
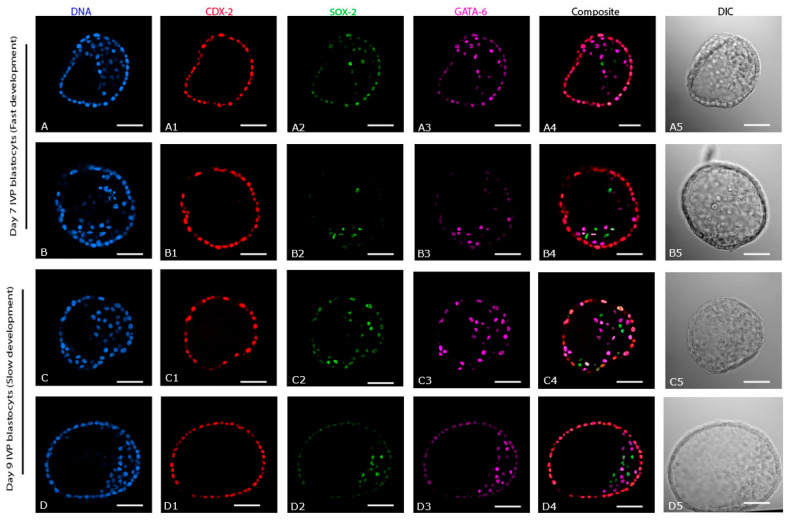
Expression of cell lineage markers (CDX-2, SOX-2, and GATA-6) in in vitro produced blastocysts; Day 7 (fast development and Day 9 (slow development). (**A**–**D**) DNA staining of all the nuclei. (**A1**–**D1**) Expression of CDX-2. (**A2**–**D2**) Expression of SOX-2. (**A3**–**D3**) Expression of GATA-6. (**A4**–**D4**) composite of CDX-2, SOX-2, and GATA-6. (**A5**–**D5**) DIC images of the respective in vitro blastocysts. Scale bar = 50 µm.

**Figure 5 ijms-24-09619-f005:**
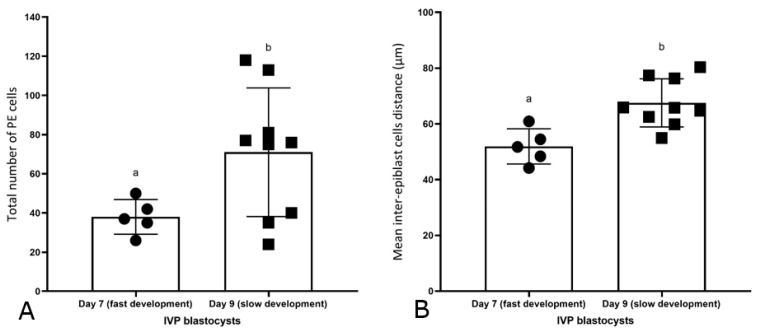
Differences between day 7 (fast-developing) and day 9 (slow developing) IVP blastocysts. (**A**) Total number of PE cells. (**B**) Mean inter-EPI cell distance. Individual values are plotted as filled squares or circles; the error bars represent the standard deviation. Different superscripts represent significant differences *p* < 0.05.

**Figure 6 ijms-24-09619-f006:**
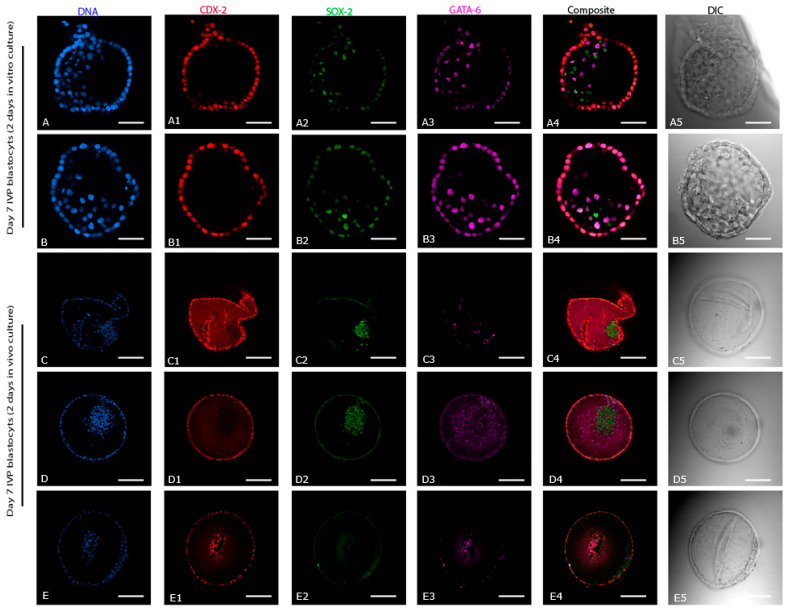
Expression of cell lineage markers (CDX-2, SOX-2, and GATA-6) in Day 7 in vitro blastocysts cultured for an additional 2 days in vitro or in vivo. (**A**–**E**) DNA staining of all the nuclei. (**A1**–**E1**) Expression of CDX-2. (**A2**–**E2**) Expression of SOX-2. (**A3**–**E3**) Expression of GATA-6. (**A4**–**E4**) Composite of CDX-2, SOX-2, and GATA-6. (**A5**–**E5**) DIC images of the respective in vitro blastocysts. Scale bar = 50 µm (**A**,**B**) and 100 µm (**C**–**E**).

## Data Availability

The data presented in this study are available within this article, and on request from the corresponding author.

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
