# Peer review of "In Vitro-Produced Equine Blastocysts Exhibit Greater Dispersal and Intermingling of Inner Cell Mass Cells than In Vivo Embryos"

_ijms, 2023, doi:10.3390/ijms24119619_

Round 1

Reviewer 1 Report

General Comments

The current study analyzed the influence of the conditions in which the embryos develop (in vivo versus in vitro), the effect of the developmental stage (early blastocyst versus blastocyst), the speed of in vitro embryo development (fast versus slow development), and environment (in vitro versus in vivo) for extended culture of in vitro derived blastocysts on the pattern of expression of cell lineages markers.

The article gives new knowledge on the developmental biology of equine species, thus important for animal reproduction and basic science.

Methods are well described; the writing is clear and follows a logical rationale. Results are sound and conclusions agreed with what is presented in the paper.

Just a couple of suggestions 

-        indicate that animal management was in compliance with the approval of a Scientific Ethics Committee

-        Line 105: describe how ovulation was confirmed on the day after insemination.

Author Response

Response to Reviewer 1 Comments

Point 1: indicate that animal management was in compliance with the approval of a Scientific Ethics Committee

Response 1: Changes have been made, Lines 101-102.

Point 2: describe how ovulation was confirmed on the day after insemination.

Response 2: Changes have been made, Line 107-109.

Reviewer 2 Report

16th May, 2023

Review of Manuscript ID: ijms- 2405654, by M. Umair et al., entitled: “In vitro-produced equine blastocysts exhibit greater dispersal and intermingling of inner cell mass cells than in vivo embryos” that is intended for publication in International Journal of Molecular Sciences

(The Microsoft Word file as Reviewer Attachment for Manuscript ID ijms-2405654 IJMS 16th May 2023 has also been added)

The Authors of the manuscript studied differences in the expression levels of markers of primarily differentiated cell lines within the 7- and 9-Day-old ICSI-derived horse blastocysts as compared to 7-Day-old in vivo-derived blastocysts recovered from artificially inseminated donor mares. 

Overall, the current work is valuable, well written in English and methodologically correct. Moreover, the wide range of results obtained is presented in an attractive graphical form. In addition, the Authors applied appropriate statistical methods to analyze the obtained results, which allowed for their insightful interpretation and critical evaluation in the context of existing knowledge.

However, in my opinion, the following minor points should have been considered prior to the acceptance of manuscript for publication as has been detailed below:

1)                  The title of manuscript has to be adjusted to the format compatible with the requirements of International Journal of Molecular Sciences.

2)                  The Conclusions section should have been enriched by the final paragraphs involving the future goals and directions of the current research according to the Reviewer recommendation indicated below:

Cumulatively, unravelling and comparatively analyzing the intra- and extra-embryonic factors that affect molecular mechanisms underlying cell commitment and inter-proteomic crosstalk between CDX-2-, SOX-2- and GATA-6-mediated intracellular hallmarks seems to be a remarkable stimulus promoting the improvement in the efficiency of generating the ex vivo-derived high-quality blastocysts in equids and other mammalian species. This also might result in the enhancement of the in vivo developmental competences after intrauterine transfer of IVP-derived embryos propagated by such advanced assisted reproductive technologies (ARTs) as in vitro fertilization (IVF) based on either gamete co-incubation or intracytoplasmic sperm injection (ICSI) [44–47] and cloning based on somatic cell nuclear transfer (SCNT) [48–50].

3)                  Taking into account the comment 1 of the Reviewer, the following 7 References have to be added according to the corresponding in-text citations included in the recommended new paragraph of the Conclusions section:

[44] Goszczynski, D.E.; Tinetti, P.S.; Choi, Y.H.; Hinrichs, K.; Ross, P.J. Genome activation in equine in vitro-produced embryos. Biol. Reprod. 2022, 106, 66–82, doi:10.1093/biolre/ioab173.

[45] Frank, B.L.; Doddman, C.D.; Stokes, J.E.; Carnevale, E.M. Association of equine oocyte and cleavage stage embryo morphology with maternal age and pregnancy after intracytoplasmic sperm injection. Reprod. Fertil. Dev. 2019, 31, 18121822, doi:10.1071/RD19250.

[46] Salamone, D.F.; Canel, N.G.; Rodríguez, M.B. Intracytoplasmic sperm injection in domestic and wild mammals. Reproduction 2017, 154, F111F124, doi:10.1530/REP-17-0357.

[47] Yao, Y.; Yang, A.; Li, G.; Wu, H.; Deng, S.; Yang, H.; Ma, W.; Lv, D.; Fu, Y.; Ji, P.; et al. Melatonin promotes the development of sheep transgenic cloned embryos by protecting donor and recipient cells. Cell Cycle 2022, 21, 13601375, doi:10.1080/15384101.2022.2051122. 

[48] Olivera, R.; Moro, L.N.; Jordan, R.; Luzzani, C.; Miriuka, S.; Radrizzani, M.; Donadeu, F.X.; Vichera, G. In Vitro and In Vivo Development of Horse Cloned Embryos Generated with iPSCs, Mesenchymal Stromal Cells and Fetal or Adult Fibroblasts as Nuclear Donors. PLoS One 2016, 11, e0164049, doi:10.1371/journal.pone.0164049.

[49] Wiater, J.; Samiec, M.; Wartalski, K.; Smorąg, Z.; Jura, J.; Słomski, R., Skrzyszowska, M.; Romek, M.; Characterization of Mono- and Bi-Transgenic Pig-Derived Epidermal Keratinocytes Expressing Human FUT2 and GLA Genes – In Vitro Studies. Int. J. Mol. Sci. 2021, 22, 9683, doi:10.3390/ijms22189683.

[50] Samiec, M.; Skrzyszowska, M. Biological transcomplementary activation as a novel and effective strategy applied to the generation of porcine somatic cell cloned embryos. Reprod. Biol. 2014, 14, 128–139, doi:10.1016/j.repbio.2013.12.006.

4)                  All the abbreviations included in the text of manuscript should have been expanded in the separate Abbreviations section added at the very end of manuscript.

Summing up, I recommend this manuscript for publication in International Journal of Molecular Sciences, provided that the above-mentioned minor remarks and suggestions pointed out by the Reviewer will have been taken into consideration by the Authors in the re-edited and resubmitted version of current paper.

The manuscript is relatively well written in English. The paper requires only fine/minor English editing.

Author Response

Response to Reviewer 1 Comments

Point 1: The title of manuscript has to be adjusted to the format compatible with the requirements of International Journal of Molecular Sciences.

Response 1: Changes have been made, Lines 2-3.

Point 2: The Conclusions section should have been enriched by the final paragraphs involving the future goals and directions of the current research according to the Reviewer recommendation indicated below:

Cumulatively, unravelling and comparatively analyzing the intra- and extra-embryonic factors that affect molecular mechanisms underlying cell commitment and inter-proteomic crosstalk between CDX-2-, SOX-2- and GATA-6-mediated intracellular hallmarks seems to be a remarkable stimulus promoting the improvement in the efficiency of generating the ex vivo-derived high-quality blastocysts in equids and other mammalian species. This also might result in the enhancement of the in vivo developmental competences after intrauterine transfer of IVP-derived embryos propagated by such advanced assisted reproductive technologies (ARTs) as in vitro fertilization (IVF) based on either gamete co-incubation or intracytoplasmic sperm injection (ICSI) [44–47] and cloning based on somatic cell nuclear transfer (SCNT) [48–50].

Response 2: Changes have been made, Lines 416-425.

Point 3: Taking into account the comment 1 of the Reviewer, the following 7 References have to be added according to the corresponding in-text citations included in the recommended new paragraph of the Conclusions section:

[44] Goszczynski, D.E.; Tinetti, P.S.; Choi, Y.H.; Hinrichs, K.; Ross, P.J. Genome activation in equine in vitro-produced embryos. Biol. Reprod. 2022, 106, 66–82, doi:10.1093/biolre/ioab173.

[45] Frank, B.L.; Doddman, C.D.; Stokes, J.E.; Carnevale, E.M. Association of equine oocyte and cleavage stage embryo morphology with maternal age and pregnancy after intracytoplasmic sperm injection. Reprod. Fertil. Dev. 2019, 31, 1812–1822, doi:10.1071/RD19250.

[46] Salamone, D.F.; Canel, N.G.; Rodríguez, M.B. Intracytoplasmic sperm injection in domestic and wild mammals. Reproduction 2017, 154, F111–F124, doi:10.1530/REP-17-0357.

[47] Yao, Y.; Yang, A.; Li, G.; Wu, H.; Deng, S.; Yang, H.; Ma, W.; Lv, D.; Fu, Y.; Ji, P.; et al. Melatonin promotes the development of sheep transgenic cloned embryos by protecting donor and recipient cells. Cell Cycle 2022, 21, 1360–1375, doi:10.1080/15384101.2022.2051122. 

[48] Olivera, R.; Moro, L.N.; Jordan, R.; Luzzani, C.; Miriuka, S.; Radrizzani, M.; Donadeu, F.X.; Vichera, G. In Vitro and In Vivo Development of Horse Cloned Embryos Generated with iPSCs, Mesenchymal Stromal Cells and Fetal or Adult Fibroblasts as Nuclear Donors. PLoS One 2016, 11, e0164049, doi:10.1371/journal.pone.0164049.

[49] Wiater, J.; Samiec, M.; Wartalski, K.; Smorąg, Z.; Jura, J.; Słomski, R., Skrzyszowska, M.; Romek, M.; Characterization of Mono- and Bi-Transgenic Pig-Derived Epidermal Keratinocytes Expressing Human FUT2 and GLA Genes – In Vitro Studies. Int. J. Mol. Sci. 2021, 22, 9683, doi:10.3390/ijms22189683.

[50] Samiec, M.; Skrzyszowska, M. Biological transcomplementary activation as a novel and effective strategy applied to the generation of porcine somatic cell cloned embryos. Reprod. Biol. 2014, 14, 128–139, doi:10.1016/j.repbio.2013.12.006.

Response 3: Changes have been made, Lines 550-567.

Point 4:All the abbreviations included in the text of manuscript should have been expanded in the separate Abbreviations section added at the very end of manuscript.

Response 4: Answer: Changes have been made, Lines 446-451.